# Lead-Free Perovskite Homojunction-Based HTM-Free Perovskite Solar Cells: Theoretical and Experimental Viewpoints

**DOI:** 10.3390/nano13060983

**Published:** 2023-03-08

**Authors:** Sajid Sajid, Salem Alzahmi, Imen Ben Salem, Jongee Park, Ihab M. Obaidat

**Affiliations:** 1Department of Chemical & Petroleum Engineering, United Arab Emirates University, Al Ain P.O. Box 15551, United Arab Emirates; s.bakhtawar@uaeu.ac.ae; 2National Water and Energy Center, United Arab Emirates University, Al Ain P.O. Box 15551, United Arab Emirates; 3College of Natural and Health Sciences, Zayed University, Abu Dhabi P.O. Box 144534, United Arab Emirates; imen.bensalem@zu.ac.ae; 4Department of Metallurgical and Materials Engineering, Atilim University, Ankara 06836, Turkey; jongee.park@atilim.edu.tr; 5Department of Physics, United Arab Emirates University, Al Ain P.O. Box 15551, United Arab Emirates

**Keywords:** lead-free, HTM-free PSC, high efficiency, simulation, experiment

## Abstract

Simplifying the design of lead-free perovskite solar cells (PSCs) has drawn a lot of interest due to their low manufacturing cost and relative non-toxic nature. Focus has been placed mostly on reducing the toxic lead element and eliminating the requirement for expensive hole transport materials (HTMs). However, in terms of power conversion efficiency (PCE), the PSCs using all charge transport materials surpass the environmentally beneficial HTM-free PSCs. The low PCEs of the lead-free HTM-free PSCs could be linked to poorer hole transport and extraction as well as lower light harvesting. In this context, a lead-free perovskite homojunction-based HTM-free PSC was investigated, and the performance was then assessed using a Solar Cell Capacitance Simulator (SCAPS). A two-step method was employed to fabricate lead-free perovskite homojunction-based HTM-free PSCs in order to validate the simulation results. The simulation results show that high hole mobility and a narrow band gap of cesium tin iodide (CsSnI_3_) boosted the hole collection and absorption spectrum, respectively. Additionally, the homojunction’s built-in electric field, which was identified using SCAPS simulations, promoted the directed transport of the photo-induced charges, lowering carrier recombination losses. Homojunction-based HTM-free PSCs having a CsSnI_3_ layer with a thickness of 100 nm, defect density of 10^15^ cm^−3^, and interface defect density of 10^18^ cm^−3^ were found to be capable of delivering high PCEs under a working temperature of 300 K. When compared to formamidinium tin iodide (FASnI_3_)-based devices, the open-circuit voltage (V_oc_), short-circuit density (J_sc_), fill factor (FF), and PCE of FASnI_3_/CsSnI_3_ homojunction-based HTM-free PSCs were all improved from 0.66 to 0.78 V, 26.07 to 27.65 mA cm^−2^, 76.37 to 79.74%, and 14.62 to 19.03%, respectively. In comparison to a FASnI_3_-based device (PCE = 8.94%), an experimentally fabricated device using homojunction of FASnI_3_/CsSnI_3_ performs better with V_oc_ of 0.84 V, J_sc_ of 22.06 mA cm^−2^, FF of 63.50%, and PCE of 11.77%. Moreover, FASnI_3_/CsSnI_3_-based PSC is more stable over time than its FASnI_3_-based counterpart, preserving 89% of its initial PCE. These findings provide promising guidelines for developing highly efficient and environmentally friendly HTM-free PSCs based on perovskite homojunction.

## 1. Introduction

Perovskite solar cells (PSCs) have attracted a lot of attention in the photovoltaic community in recent years owing to their high power conversion efficiencies (PCEs) and low-cost fabrication [1,2]. Additionally, they have shown astounding improvements in photoelectric performance, which are a result of the superior optoelectronic characteristics of organic–inorganic halide perovskites, including their exceptionally high absorption coefficient, suitable band gap, ambipolar carrier transport property, and high defect tolerance. The PCEs of PSCs have lately approached 25.7% [3], which is a significant improvement. The commercialization of the PSCs is, however, limited by the use of costly organic hole transporting materials (HTMs) [4,5,6,7,8] and the toxicity of the lead (Pb) element [9]. The HTM-free PSCs are thought to significantly reduce the fabrication cost in this scenario. Consequently, a great deal of innovative studies focused on the HTM-free PSCs due to their inexpensive production costs and straightforward processing methods. The first methylammonium lead iodide and titanium oxide (MAPbI_3_/TiO_2_) heterojunction-based HTM-free PSCs showed PCEs as high as 7.28% [10]. Mei et al. reported PSCs with a heterojunction based on 5-ammoniumvaleric acid/methylammonium lead iodide/zirconium oxide/titanium oxide [(5-AVA)_x_(MA)_1−x_PbI_3_/ZrO_2_/TiO_2_)], which yielded a certified PCE of 12.8% [11]. A PCE of 14.38% in MAPbI_3_ heterojunction-based HTM-free PSCs was obtained through further investigation [12]. Huang developed HTM-free PSCs using 2,3,5,6-tetrafluoro-7,7,8,8-tetracyanoquinodimethane (F4TCNQ)-doped MAPbI_3_ and demonstrated a PCE of 18.85% [13]. Kong et al. further reported one efficient method to increase the PCE of heterojunction-based HTM-free PSCs, and the PCE reached 19.42% [14]. Difluorobenzylamine has recently been shown to function as an interfacial modifier to stabilize and improve the efficiency of HTM-free PSCs. Density-functional theory calculations and experiments showed that the fluorine atoms in the benzene ring can passivate defects at the surface or interface of the perovskites, enhancing the transfer of charge-carriers. As a result, PCE of the HTM-free PSC reached 14.6%, and after 1680 h of storage at 20–30% air humidity, the PCE of the unencapsulated device was still 92% of the initial PCE [15]. However, the reported PCEs of HTM-free PSCs hardly even reach 20%. In addition, these HTM-free PSCs contain toxic Pb, which limits their commercialization.

Tin (Sn)-based perovskites have garnered increased attention in the photovoltaics field due to their environmental friendliness compared to Pb-based PSCs [16]. Sn is the first and most likely alternative in the quest to replace lead in PSCs because, in comparison to Pb, it is less hazardous under various test conditions [17]. Sn-based perovskites have the potential to outperform their lead-based counterparts from a physical and theoretical perspective due to their lower bandgap, which is closer to the ideal values established by the Shockley–Queisser limit [18], high carrier mobility, and low exciton binding energy. Lee et al. reported that FASnI_3_-based PSCs had a 4.8% PCE and good repeatability [19]. Later, Liao et al. showed an inverted PSC with a PCE of 6.22% [20]. The PCEs were further improved with pure FASnI_3_ perovskite, up to 7 to 8% [21,22,23,24]. Higher PCEs might be attained by partially substituting other organic cations for formamidinium (FA). For instance, 8 to 9% PCEs have been obtained for enhanced film morphology and decreased carrier recombination by substituting 25% of the FA with methylammonium [25,26]. Over 9% PCE was achieved when bulky organic cations, such as phenylethylammonium, were added to the FASnI_3_ perovskite lattice to generate a two-dimensional/three-dimensional hybrid perovskite film [27,28]. For Sn-based PSCs, a PCE of 9.6% was achieved by combining mixed guanidium and FA cations with the addition of ethylenediammonium diiodide [29]. More recently, after being post-treated with edamine Lewis base, a 10.18% PCE with FA_0.98_EDA_0.01_SnI_3_ perovskite film has been reported [30]. Ning et al. [31] have recently created tin iodide/dimethyl sulfoxide (SnI_2_.(DMSO)_x_) complexes with improved coordination using phenethylamine bromide (PEABr) as an additive by utilizing the interaction between I_2_ and DMSO. In contrast, it was discovered that SnI_2_ may be evenly distributed into the precursor solution and lessens the quantity of uncoordinated SnI_2_. Obtaining more uniform perovskite films while guiding out-of-plane crystal orientation is possible with highly coordinated SnI_2_.(DMSO)_x_. The electron diffusion length in Sn-based film increased to roughly 80 nm and the PCE of 14.6% was obtained in PSCs as compared to reference films. Studies using SCAPS simulations have also shown that FASnI_3_-based PSCs can outperform lead-based ones [32]. Though great progress has been made in FASnI_3_- or MASnI_3_-based devices, the presence of organic ions in FASnI_3_ poses some concerns about the durability of PSCs based on organic–inorganic hybrid Sn-based perovskites due to the volatile nature of organic constituents. Despite the poor morphology of their films, all-inorganic Sn-based perovskite may possibly be a better choice in this scenario. Since CsSnX_3_ (here X represent halides: I, Br, or Cl) perovskites have tolerance factors that are closer to 1, the geometric arrangement of CsSnX_3_ is more durable [33]. The CsSnX_3_ perovskites have low energies for the phase transition from the orthorhombic to the tetragonal phase and, as a result, low phase transition temperatures [34]. Furthermore, the narrow band gap (1.3 eV), high absorption coefficient (10^4^ cm^−1^), and high hole mobility (42 cm^2^ V^−1^ s^−1^) of CsSnI_3_ make it an excellent perovskite for high optical absorption and charge-carrier transport [35,36,37,38]. In comparison to the theoretical limit, the PCE of reported PSCs based on CsSnI_3_ is only about 5.03% [39]. However, it is important to consider the oxidation of Sn^2+^ to Sn^4+^ because this will induce perovskite instability and decrease the efficiency of Sn-based PSCs [40]. In this context, metallic Sn was used by Lin et al. to reduce the Sn^2+^ from oxidation [41]. Other ways have been described, such as adding chemicals to the perovskite precursor, graphene–Sn quantum dots [42], tin halide (SnF_2_/SnCl_2_) [43,44,45], hypophosphorous [46], hydrazine [47], ethylenediammonium [48], and 1,4-bis(trimethylsilyl)-2 [49]. Therefore, such approaches might increase the Sn-based PSCs’ stability, which would then increase their performance. However, the HTM-free PSCs seldom ever use the environmentally favorable homojunction-based perovskites. The perovskite and electron/hole transport materials form a heterojunction, but because of its excellent self-doping ability [50,51,52,53], the perovskite can also form a homojunction, which would lessen the presence of impurities that could act as carrier recombination centers [54]. The guided transport of the photo-induced electrons and holes may be aided by an internal electric field that arises in the perovskite homojunction, which would reduce additional charge-carrier recombination in the perovskite layer [55]. It is therefore a promising route to further improve PSC performances. If homojunction Sn-based perovskites might be theoretically and experimentally used to guide the fabrication of efficient HTM-free PSCs, it is currently unclear and scarcely reported.

As FASnI_3_ and CsSnI_3_ have similar perovskite crystal structures, fabricating CsSnI_3_ on FASnI_3_ may produce a homojunction layer that is high quality and has fewer defects. In light of this, conducting thorough theoretical and experimental examinations of the performance of FASnI_3_/CsSnI_3_ homojunction-based HTM-free PSCs could be a promising strategy to offer guidelines for environmentally friendly devices. In this regard, using SCAPS software, we first investigated the parameters of p-type CsSnI_3_ that influence the performance of lead-free homojunction-based HTM-free PSCs. The HTM-free PSCs in this study use a lead-free homojunction-based perovskite (FASnI_3_/CsSnI_3_) as the photon harvesting layer. In the light absorption spectrum, the homojunction-based FASnI_3_/CsSnI_3_ perovskite demonstrated better photon harvesting. Additionally, the homojunction’s built-in electric field, which was identified using SCAPS simulations, promotes the directed movement of the photo-induced charges, lowering carrier recombination losses. With optimal conditions, the lead-free homojunction-based HTM-free PSCs display a PCE of 19.02%. Moreover, a homojunction-based HTM-free PSC was fabricated to verify the validity of the simulations. According to the experimental approach, the FASnI_3_/CsSnI_3_ perovskite-based homojunction was fabricated by a two-step process [55]. As a result, the FASnI_3_/CsSnI_3_ homojunction-based HTM-free device demonstrated better performance with a PCE of 11.77% compared to the FASnI_3_-based HTM-free PSC (PCE = 8.94%).

## 2. Materials and Methods

### 2.1. Simulation Procedure

For experimental scholars investigating solar cells, finding the ideal structure and materials have always been a major difficulty. This procedure can be expedited by the simulation tool. The unique capabilities of the SCAPS include, but are not restricted to, modeling up to seven layers and computing numerous parameters, such as spectral response, energy bands, J-V curve, and defect density, by only resolving three fundamental semiconductor equations. It is simple to operate and can be utilized in both light and dark conditions. Key processes in photovoltaic systems are defined by SCAPS modeling techniques, allowing for the intuitive and systematic classification of optimal operating conditions for each parameter. In this research, a lead-free homojunction-based perovskite with a device layout of FTO/TiO_2_/FASnI_3_/CsSnI_3_/Au was proposed in HTM-free PSCs, as illustrated in Figure 1a,b. The illumination source in the simulation was the AM l.5G solar radiation spectrum with an intensity of 1000 W·m^−2^. Important material parameters were gathered from prior experimental reports and are summarized in Table 1 to validate the viability of the simulations. Two interface layers were introduced: one between the FASnI_3_ and CsSnI_3_ and the other one between the CsSnI_3_ and gold (Au) electrode, accounting for the interfacial defect density (denoted by IDL1 and IDL2 in Table 1). The defects in the perovskite layer were configured to have a characteristic energy of 0.1 eV and to be in a neutral Gaussian distribution with energy 0.6 eV above the valence band. Interfacial defects were regarded as neutral single defects with distribution energy of 0.6 eV above the valence band.

Using SCAPS-1D software developed by researchers at the University of Ghent, the photovoltaic performance of the devices under AM1.5G solar illumination was examined [68]. By resolving the following Equations (1)–(3), which are encoded in SCAPS-1D, it was possible to obtain the current-voltage (J-V) characteristic curves, external quantum efficiency (EQE), generation/recombination of charges, and electric field distribution.
(1)$$\frac{d}{dx}\left(-ɛ\left(x\right)\frac{d\Psi }{dx}\right)=q\left[p\left(x\right)-n\left(x\right)+N_{d}^{+}\left(x\right)-N_{a}^{-}\left(x\right)+P_{t}\left(x\right)-n_{t}\left(x\right)\right]$$
(2)$$\frac{dP_{n}}{dt}=G_{p}-\frac{P_{n}-P_{n0}}{\tau_{p}}-P_{n}{µ}_{n}\frac{dE}{dx}-{µ}_{p}E\frac{dP_{n}}{dx}+D_{p}\frac{d^{2}P_{n}}{dx^{2}}$$
(3)$$\frac{dn_{p}}{dt}=G_{n}-\frac{n_{p}-n_{p0}}{\tau_{n}}-n_{p}{µ}_{p}\frac{dE}{dx}-{µ}_{n}E\frac{dn_{p}}{dx}+D_{n}\frac{d^{2}n_{p}}{dx^{2}}$$
where *ɛ*, *Ψ*, *q*, *n*, *p*, *n_t_*, *p_t_*, *N_d_*, *N_a_*, *G*, *D*, and *E* represent the permittivity, electrostatic potential, elementary charge, density of free electrons, density of free holes, density of trapped electrons, density of trapped holes, donor doping density, acceptor doping density, generation rate, diffusion coefficient, and electric field, respectively.

### 2.2. Experimental Procedure

#### 2.2.1. Fabrication of PSCs

The FTO-substrates were ultrasonically cleaned after being immersed for 20 min in acetone, deionized water, detergent solution, and isopropyl alcohol, respectively. The FTO-substrates were treated with UV-ozone for 15 min following nitrogen blow drying. An aqueous stock solution of 2 M TiCl_4_ (stored in the freezer) was diluted to the desired proportion after cleaning the FTO-substrates. The as-cleaned substrates were then dipped in this solution and maintained in a closed vessel in a 70 °C oven for one hour. To create a compact layer of TiO_2_ (denoted as c-TiO_2_), the substrates were dried at 100 °C in the air for an hour after being rinsed with deionized water and ethanol. Spin-coating the TiO_2_ nanoparticles paste in ethanol at 2000 rpm for 25 s resulted in the deposition of a mesoscopic TiO_2_ layer (denoted as m-TiO_2_). The substrate was subsequently annealed for 20 min at 500 °C. After annealing, a diluted TiCl_4_ aqueous solution in deionized water was applied to the FTO/c-TiO_2_/m-TiO_2_ substrate for 20 min at 70 °C. The substrate was then exposed to UV-ozone for 10 min. SnI_2_ (0.143 g) and FAI (0.140 g) were combined with 1 mL of DMSO to form the precursor solution for the FASnI_3_. The FASnI_3_ layer was then created by one-step spin-coating of the FASnI_3_ precursor solution for 25 s at 4000 rpm. Then, 0.5 mL of diethyl ether was carefully injected during the spinning process as an anti-solvent to speed up the formation of perovskite film. The resultant perovskite layers were annealed for 10 s at 60 °C and then for 12 min at 100 °C. Then, the FTO/c-TiO_2_/m-TiO_2_/FASnI_3_ coated substrates were used to fabricate a homojunction FASnI_3_/CsSnI_3_-based perovskite layer, as schematically illustrated in Figure 2. With a SnI_2_ thickness of ~65 nm (formed by vapor deposition) and a dipping duration of 50 s (equivalent to a SnI_2_/CsI ratio of 1.6), the top CsSnI_3_ perovskite layer was produced. The as-prepared FTO/TiO_2_/FASnI_3_/CsSnI_3_ substrate was then annealed at 70 °C for 10 min. Note that we followed the two-step method from a previous report [55]. Lastly, to fabricate 80 nm electrodes onto the perovskite films and complete the FTO/c-TiO_2_/m-TiO_2_/FASnI_3_/Au and FTO/c-TiO_2_/m-TiO_2_/FASnI_3_/CsSnI_3_/Au devices, gold was thermally evaporated.

#### 2.2.2. Characterization of Thin-Films and PSCs

The surface and cross-sectional morphologies were examined using a scanning electron microscope (Hitachi S-4800). The absorption spectrum was determined using a UV–Vis spectrophotometer (UV-2600). Using Edinburg PLS 980, the steady-state PL spectra of the obtained samples were examined. The TRPL decay of the perovskite layers was shown using a transient state spectrophotometer (Edinburgh Institute F900) and a 485 nm laser. For the J-V characteristic curves, the devices were measured using a source meter (Keithley 2400) using forward (−0.1 to 1.2 V) or reverse (1.2 to −0.1 V) scans from a solar simulator (XES-301S+EL-100) under AM 1.5G illumination with a power intensity of 1000 W cm^−2^. The step voltage was set to 12 mV, and the delay duration was 10 ms. The QE-R system (Enli Tech.) was used to determine the EQEs of the as-prepared devices.

## 3. Results and Discussion

We first explore significant parameters (such as thickness, doping density, and interface defect density at IDL1 and IDL2, and defect density of the CsSnI_3_) that affect the performance of the homojunction-based HTM-free PSCs in order to acquire optimum values and provide guidelines for the experimental approaches. These parameters were left the same for the remaining layers of the as-simulated HTM-free PSCs. The performance variances in the cells as a function of the aforementioned parameters are elaborated in the following discussion.

Given that the thickness of the CsSnI_3_ has a significant role in the incident light absorption and charge-carrier generation, the PSC’s performance will be affected by varying this thickness. The perovskite layer in a PSC must be thick enough to maximize light absorption yet thick enough to boost the collection of photo-generated charge-carriers. The collection efficiency is determined by the competition between recombination and charge transfer to the corresponding contacts. Therefore, choosing the right CsSnI_3_ thickness will be crucial for developing efficient homojunction-based HTM-free PSCs. Figure 3 illustrates how the photovoltaic performance and energy levels of the as-simulated HTM-free PSCs vary with CsSnI_3_ thickness. As noted in Figure 3a, the J_sc_ was increased significantly from 27.08 to 30.09 mA cm^−2^, which could be attributed to a higher generation of charge-carriers due to an improved capability of the homojunction-based perovskite layer for light harvesting. According to the simulated quantum efficiency, as shown in Figure 3b, the absorption spectrum broadens with increasing CsSnI_3_ thickness. It should be noted that the J_sc_ grows slowly and tends to saturate when the CsSnI_3_ thickness exceeds 300 nm because too thick perovskite could result in a significant charge-carrier recombination [64]. Figure 3c shows how the perovskite layer tends to saturate the PCE at larger thicknesses. This is because thick layers make it hard for charge-carriers to be collected in a timely manner, and they recombine before they reach the relevant electrodes. Even though the simulation results showed that PSCs with thicker CsSnI_3_ layers (more than 100 nm and less than 400 nm) performed better, we still believed that it would be difficult to fabricate CsSnI_3_ layers on FASnI_3_ that were thicker than 100 nm because it is known that the precursor solvent used to make CsSnI_3_ will dissolve the FASnI_3_ bottom layer. Figure 3d shows the energy levels of the HTM-free PSCs as a function of CsSnI_3_ thickness, demonstrating that the built-in electric potential (*V_bi_*) is weakly influenced by the thickness of CsSnI_3_. This is because the doping density, as shown by Equation (4), is the primary factor controlling the *V_bi_* [69,70].
(4)$$V_{bi}=\frac{kT}{q}ln\frac{N_{A}N_{D}}{n_{i}^{2}}$$
where *k*, *T*, *q*, *N_A_*, *N_D_*, and *n_i_* represent Boltzmann constant, temperature, elementary charge, acceptor doping density, donor doping density, and intrinsic density, respectively. These findings reveal that when the light-harvesting perovskite layer is very thin, poor photon absorption results in low photocurrents; however, if the perovskite layer is very thick, recombination issues develop due to poor charge-carrier extraction efficiency. The simulation findings showed that a CsSnI_3_ layer with an optimal thickness of 100 nm can be used to obtain efficient homojunction-based HTM-free PSCs.

Another crucial element that influences the PSC’s photovoltaic metrics is the doping density of the perovskite layer. We therefore look at the critical role that the CsSnI_3_ doping density plays in the homojunction-based HTM-free PSCs. Appendix A shows the alterations in the J-V characteristic curves, PCEs, and energy levels with changes in the acceptor doping density (*N_A_*). Appendix A shows that no discernible changes in the photovoltaic parameters were seen at *N_A_* from 10^13^ to 10^15^ cm^−3^, demonstrating that the relatively modest doping density does not alter the inherent property of CsSnI_3_. The better photovoltaic performance of the as-simulated HTM-free PSCs can be due to the larger *V_bi_* (as shown in Appendix A and Equation (4)) brought on by the larger *N_A_*, which is increased from 10^15^ to 10^18^ cm^−3^. As shown in Appendix A, the PCE improves when the *N_A_* increases from 10^15^ to 10^18^ cm^−3^ and it begins to decline as the *N_A_* of the CsSnI_3_ exceeds 10^18^ cm^−3^. High doping density can help with charge-carrier separation, but too much doping inhibits charge-carrier mobility and causes substantial charge-carrier recombination, which lowers PCE. The simulation results reveal that the optimal value of *N_A_* is close to 10^18^ cm^−3^.

The perovskite layer is the main part for the generation of charge-carriers. As a result, the PSC’s performance is greatly influenced by the properties of the perovskite layer. Defect density is a crucial indicator of how well the perovskite layer performs. Therefore, fabricating high-quality perovskite active layers with low defect densities and less nonradiative recombination is essential for producing highly efficient PSCs. The nature and density of defect states in perovskites are very sensitive to the film deposition conditions, according to prior calculations and experimental data [71]. Due to the lack of stoichiometric compositions at the surfaces of grains and the possibility that perovskite constituents could disintegrate during the thermal annealing process and leave defects, it is believed that the majority of the defects in perovskite films are found at the grain boundaries or the interface [54,72]. In this context, we have investigated the impact of defect densities on the performance of the homojunction-based HTM-free PSCs, specifically the effect of bulk defects in the perovskite layer and defects at the interfaces. Figure 4a displays the J-V characteristic curves as function of defect density in the CsSnI_3_. It demonstrates that the PCE begins to decline when the perovskite layer has a high defect density. Higher defect densities in low-quality perovskite layers raise carrier recombination rates because defects trap charge-carriers, as seen in Figure 4b. It is currently challenging to significantly reduce the defect density in the CsSnI_3_ layer due to the fact that the CsSnI_3_ perovskite is unstable in the air and requires sophisticated fabrication techniques [36,45,73]. According to the simulation results, high device performance can be obtained with defect density from 10^15^ to 10^16^ cm^−3^ in the CsSnI_3_ layer.

In addition, the performance of PSCs is significantly influenced by the characteristics of the interface layers. It is widely speculated that low-quality interfaces result in larger surface/interface defect densities, which in turn increase the recombination of charge-carriers [74]. We assume that the defect densities of IDL1 and IDL2 range between 10^18^ and 10^22^ cm^−3^ in order to discuss the impact of the defect density of the interface layers on the PSC performance. The J-V characteristic curves and PCEs of the as-simulated homojunction-based HTM-free PSCs as a function of defect densities in IDL1 and IDL2 are shown in Appendix A. These graphs demonstrate that lowering defect density in the interface layer will boost the PCE. Evidently, the effect of the defect density on the J-V characteristics becomes noticeable when the defect density is more than 10^18^ cm^−3^, which deteriorates the performance of the homojunction-based HTM-free PSCs. It is clear that IDL1 has a higher impact on PCE than IDL2, which can be explained by the possibility that defects may occur during the deposition of the CsSnI_3_ layer on FASnI_3_ thin-film. Based on simulation results, the defect densities of IDL1 and IDL2 should be less than 10^19^ cm^−3^, in order to attain the desired efficiency of homojunction-based HTM-free PSCs.

The homojunction-based FASnI_3_/CsSnI_3_ is utilized in this study as an absorber layer in the lead-free HTM-free PSCs. It is revealed that the thickness, doping density, and defect density of the CsSnI_3_ layer should be around 100 nm, 10^17^ cm^−3^, and 10^16^ cm^−3^, respectively, for the HTM-free PSCs to perform at their best. The optimal performance of the as-simulated HTM-free PSCs is shown in Figure 5 and the photovoltaic parameters are listed in Table 2. The improvement of the overall performance of the homojunction-based HTM-free PSC (Figure 5a) is attributed to the increased photo-generated charge-carrier generation rate (Figure 5b) and extended absorption of the homojunction-based FASnI_3_/CsSnI_3_ (Figure 5c). The electric field within the PSC might be impacted by unintentional defects in the bulk or at device interfaces. As shown in Figure 5d, the electric field generated by the FASnI_3_/CsSnI_3_ homojunction with a defect density of 10^16^ cm^−3^ and doping density of 10^17^ cm^−3^ enables directed charge-carrier transfer. Free holes will rise as electron concentrations decrease in the p-type perovskite region, exhibiting carrier-directed transport [55]. It should be noted that when the doping density is equal to or lower than the defect density, the perovskite layers become semi-insulating and fail to form the desired p-n homojunction [75]. Based on the simulation results, the PCE of FASnI_3_/CsSnI_3_ homojunction-based HTM-free PSCs significantly increased from 14.62 to 19.03% when compared to the devices based on a bare FASnI_3_ perovskite layer.

Meanwhile, the performance of the as-simulated HTM-free PSCs is tested as a function of operating temperature. For PSC simulation, 300 K is typically the testing temperature used. However, under experimental conditions, the operational temperature is typically higher than the simulation. Here, under the conditions stated in Table 1, the working temperature for the efficient homojunction-based HTM-free PSC was maintained at 300 K. It is important to note that high temperatures may cause devices to deform and undergo higher stress [76], which will increase interfacial defects and poor interconnectivity between the layers, lowering the photovoltaic parameters, as depicted in Appendix A. Despite the modeling findings demonstrating reduction in the photovoltaic parameters of the as-simulated HTM-free PSCs, the stability of the Sn-based PSCs can be increased, though, by modifying the active layer morphology [77,78], using a controlled environment for film formation [36,47], and adding tin halides (SnF_2_ or SnCl_2_) to the perovskite precursor solution [79]. According to some studies, FASnI_3_ and CsSnI_3_ exhibit significantly better thermal and photochemical stability with regard to both light and high temperatures [45,80,81].

The performance of the homojunction-based PSCs was assessed, and then FASnI_3_ and FASnI_3_/CsSnI_3_ perovskite thin-films were experimentally fabricated. To make the FASnI_3_ thin-film, a one-step spin-coating was taken. Contrarily, a two-step technique was used to fabricate the CsSnI_3_ layer on top of a FASnI_3_ layer (made through a one-step spin-coating process), as shown in Figure 2. Both perovskite (FASnI_3_ and FASnI_3_/CsSnI_3_) layers are compact and pinhole-free, as shown by surface morphologies and cross-sectional SEM images (Figure 6a–d), which will aid in charge-carrier transport and prevent shunting paths. Figure 6e shows the optical absorption of the FASnI_3_ and FASnI_3_/CsSnI_3_ thin-films. According to the results, the FASnI_3_/CsSnI_3_ sample exhibits improved optical absorption, which is advantageous for the generation of photo-induced charge-carriers in the homojunction-based HTM-free PSCs. The charge-carrier dynamics within as-prepared samples were investigated by steady-state photoluminescence (PL) and time-resolved photoluminescence (TRPL) spectroscopies as depicted in Figure 6. Compared to FASnI_3_, the PL of the FASnI_3_/CsSnI_3_ sample decreases, as shown in Figure 6f. The higher quenched PL spectra shows that in the final device, the FASnI_3_/CsSnI_3_ film has a substantially higher charge extraction and transfer efficiency. The FASnI_3_/CsSnI_3_ sample has a shorter decay lifespan than the FASnI_3_ sample, according to the TRPL spectra (Figure 6g), indicating that the charge-carriers were extracted quickly at the FASnI_3_/CsSnI_3_ interface. In order to maximize the efficiency of charge-carrier transport within homojunction-based HTM-free PSCs, the FASnI_3_/CsSnI_3_ layer could reduce charge losses.

We fabricated two different types of PSCs (FTO/c-TiO_2_/m-TiO_2_/FASnI_3_/Au and FTO/c-TiO_2_/m-TiO_2_/FASnI_3_/CsSnI_3_/Au) while taking into consideration the aforementioned simulation results and optoelectronic characteristics of the perovskite thin-films. Figure 7a,b shows the cross-sectional SEM images of the PSCs prepared using FASnI_3_ and FASnI_3_/CsSnI_3_ layers. The surface and cross-sectional SEM results show that the functional layers in both PSCs were compact and free of pinholes. The photovoltaic parameters are presented in Table 3. Figure 7c displays the J-V characteristic curves for the FASnI_3_ and FASnI_3_/CsSnI_3_ devices under AM1.5G illumination. The average PCE of the FASnI_3_/CsSnI_3_ HTM-free PSC was 11.77%, with a V_oc_ of 0.84 V, a J_sc_ of 22.06 mA cm^−2^, and an FF of 63.50%. In comparison, the PCE of the FASnI_3_ device was significantly lower with PCE of 8.94%, V_oc_ of 0.79 V, J_sc_ of 21.20 mA cm^−2^, and FF of 53.35%. The efficient extraction of charge-carriers and improved light harvesting are attributable for the improved performance of FASnI_3_/CsSnI_3_-based PSCs, as depicted in Figure 7d. The simulation results showed higher PCEs than those for experimentally prepared HTM-free PSCs, which we believed might be due to the defects in the perovskite layer. In this scenario, it is possible to further improve performance of the homojunction-based HTM-free PSCs by incorporating various additives into the perovskite precursor, such as N,N’-methylenebis(acrylamide) and pentafluorophen-oxyethylammonium iodide [82,83]. The diffusion of the Au metal, which may create serious deep centers, could be another possible cause of the inferior performance of the as-prepared HTM-free PSCs. In this context, we suggest interfacial engineering to promote favorable charge transport in perovskites [84]. Another potential method to increase the efficacy of the HTM-free PSCs is to introduce a plasmonic effect, which has been shown to promote the optoelectronic characteristics of the perovskites [85,86].

In addition, the statistical data of PSCs based on FASnI_3_ and FASnI_3_/CsSnI_3_ (33 devices for each) were obtained, as shown in Appendix A. The statistical data show that HTM-free PSCs prepared with the FASnI_3_/CsSnI_3_ homojunction have higher PCEs and better reproducibility than the FASnI_3_-based HTM-free devices. The PCEs for FASnI_3_/CsSnI_3_ homojunction-based HTM-free PSCs fall within a narrow range, whereas FASnI_3_-based devices exhibit PCEs that are widely distributed. The average PCE for the FASnI_3_-based devices reached 5.78%. The average PCE for FASnI_3_/CsSnI_3_ homojunction-based HTM-free PSCs, in comparison, reaches up to 10.83%. The high PCE and reproducibility of the HTM-free PSCs are two important characteristics, but another is their stability. The steady-state J_sc_ and PCE of the FASnI_3_/CsSnI_3_-based HTM-free PSCs were initially tracked at a maximum power point at a bias voltage of 0.9 V at room temperature under AM 1.5G illumination, as shown in Appendix A. After 140 s of continuous illumination, the FASnI_3_/CsSnI_3_-based HTM-free PSC exhibits a stabilized PCE of 11.58%, which is considerably near to the maximum PCE of 11.77% for the identical PSC. Further, unsealed PSCs made using FASnI_3_ and FASnI_3_/CsSnI_3_ perovskites had their long-term stability tested every 45 min for 500 min under AM 1.5G illumination, as depicted in Appendix A. After each stability test, the measured devices were kept in a glass oven at 25 °C and 55% humidity. Appendix A illustrates the superior stability of the FASnI_3_/CsSnI_3_-based HTM-free PSCs over FASnI_3_-based devices. After 500 min of testing, the efficiency of FASnI_3_/CsSnI_3_-based HTM-free PSCs slightly decreases from 11.73 to 10.43% while maintaining 89% of the original PCE. The PCE of FASnI_3_-based PSCs under the identical testing conditions declined sharply from 8.58 to 1.04%, demonstrating poor stability and quick degradation. The protection of the CsSnI_3_ layer, which lessens the interaction of degradation factors such as moisture and oxygen present in the environment, can be attributed to the higher stability of FASnI_3_/CsSnI_3_-based devices.

## 4. Conclusions

In this research, numerical analysis was first used to determine the optimum conditions for fabricating highly efficient homojunction-based HTM-free PSCs. There were a number of factors that have all been carefully considered and the influences on the performance of the HTM-free PSCs were studied. The simulation results revealed the broad absorption spectrum by the inclusion of CsSnI_3_, and the high hole mobility facilitated the charge-carrier transportation. For effective photon harvesting in highly efficient HTM-free PSCs, a CsSnI_3_ layer thickness of approximately 100 nm was required. The optimal doping density for the CsSnI_3_ layer was between 10^15^ and 10^18^ cm^−3^. The simulation results also showed that the PSC operates better when its defect density was suitable, but that it performs poorly and has a greater charge-carrier recombination rate when it has too many defects. The simulation method indicated that the FASnI_3_/CsSnI_3_ homojunction-based HTM-free PSCs displayed PCEs up to 19.03% under optimal conditions. Taking into account the aforesaid simulation results, an experimental FASnI_3_/CsSnI_3_ homojunction-based HTM-free PSC with a configuration of FTO/c-TiO_2_/m-TiO_2_/FASnI_3_/CsSnI_3_/Au was fabricated. Compared to the FASnI_3_/CsSnI_3_-based HTM-free PSC, which delivered a PCE of 11.77%, the PCE of the FASnI_3_ device was much lower (8.94%). The FASnI_3_/CsSnI_3_-based HTM-free PSC additionally showed long-term stability while retaining 89% of its initial efficiency. Our study is crucial for providing new viewpoints on using lead-free homojunction perovskites and for further enhancing the performance of lead-free homojunction-based HTM-free PSCs.

## Figures and Tables

**Figure 1 nanomaterials-13-00983-f001:**
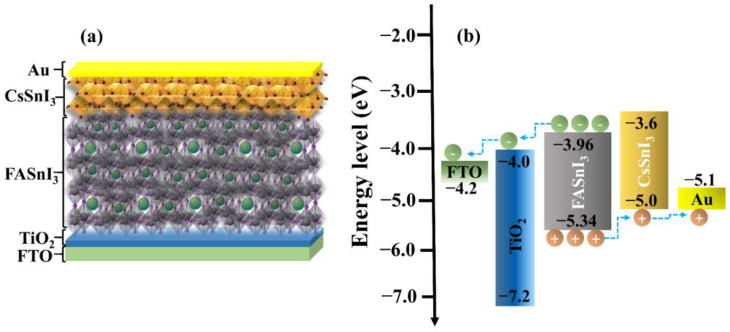
Schematic illustrations of a homojunction-based HTM-free PSC (**a**) and a corresponding energy band diagram of each material used in the simulation (**b**).

**Figure 2 nanomaterials-13-00983-f002:**
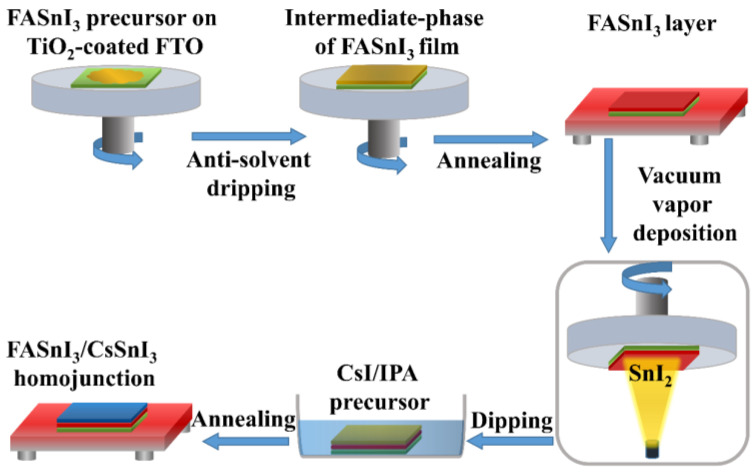
The illustration for the fabrication of the FASnI_3_/CsSnI_3_ homojunction-based layer.

**Figure 3 nanomaterials-13-00983-f003:**
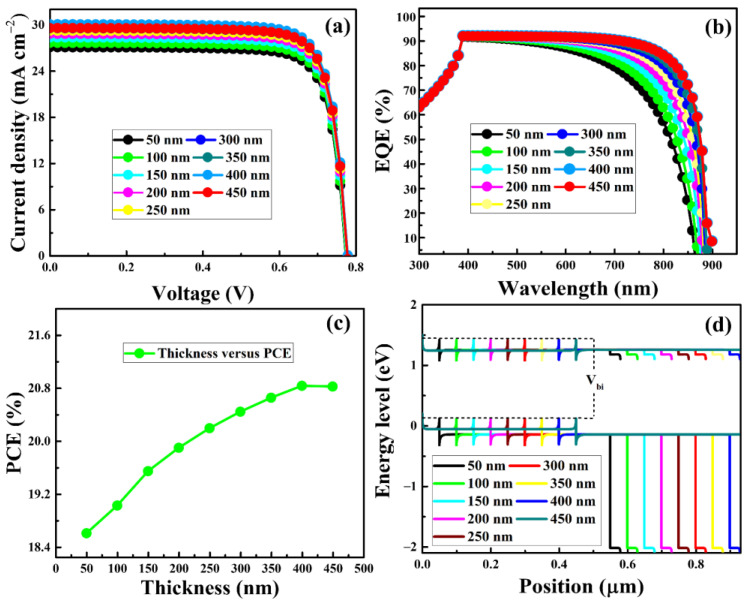
Performance of the homojunction-based HTM-free PSCs as a function of perovskite (CsSnI_3_) layer thickness. The current-voltage (J-V) characteristic curves (**a**), external quantum efficiencies (EQEs) (**b**), power conversion efficiencies (PCEs) (**c**), and energy levels (**d**).

**Figure 4 nanomaterials-13-00983-f004:**
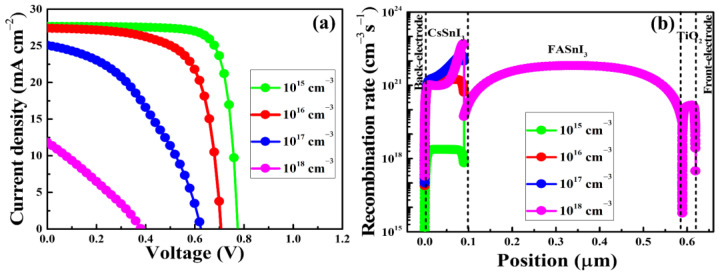
The current-voltage (J-V) characteristic curves (**a**) and charge-carrier recombination rate (**b**) of the as simulated homojunction-based HTM-free PSCs as a function of defect density of CsSnI_3_.

**Figure 5 nanomaterials-13-00983-f005:**
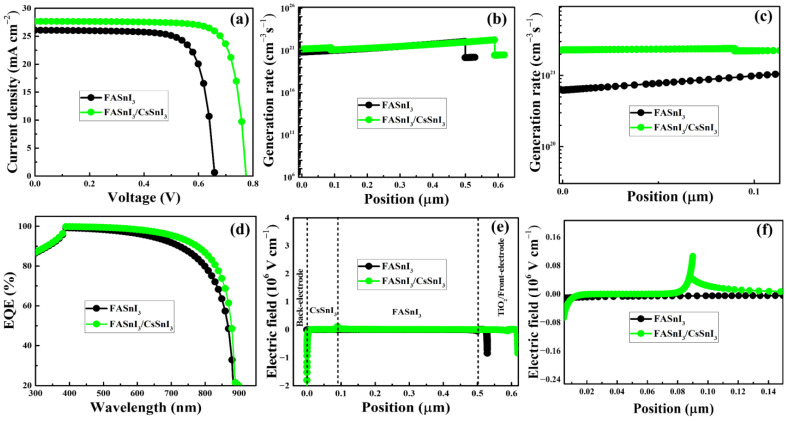
Photovoltaic performance of the HTM-free PSCs utilizing bare FASnI_3_ and homojunction-FASnI_3_/CsSnI_3_ perovskite. The J-V characteristic curves (**a**), charge-carrier generation rate (**b**), zoomed-in graph showing the improved charge-carrier generation (**c**), external quantum efficiencies (EQEs) (**d**), and electric field distribution (**e**), zoomed-in graph, indicating enhanced electric field (**f**).

**Figure 6 nanomaterials-13-00983-f006:**
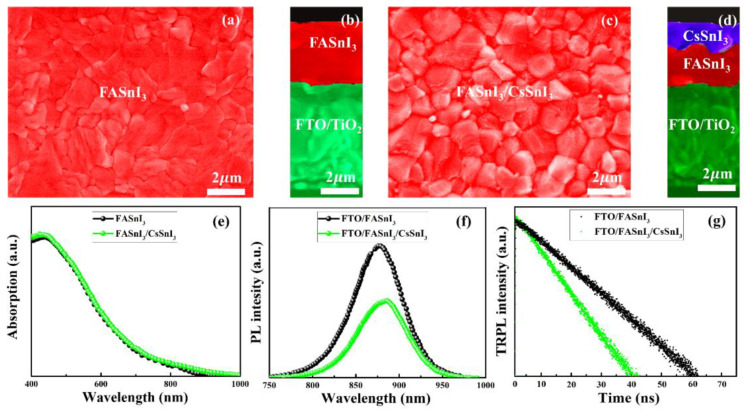
Surface and cross-sectional SEM images of FASnI_3_ thin-films (**a**,**b**) and FASnI_3_/CsSnI_3_ thin-films (**c**,**d**). UV–Vis absorption spectra (**e**), PL spectra, and TRPL spectra of the as-prepared samples (**f**,**g**).

**Figure 7 nanomaterials-13-00983-f007:**
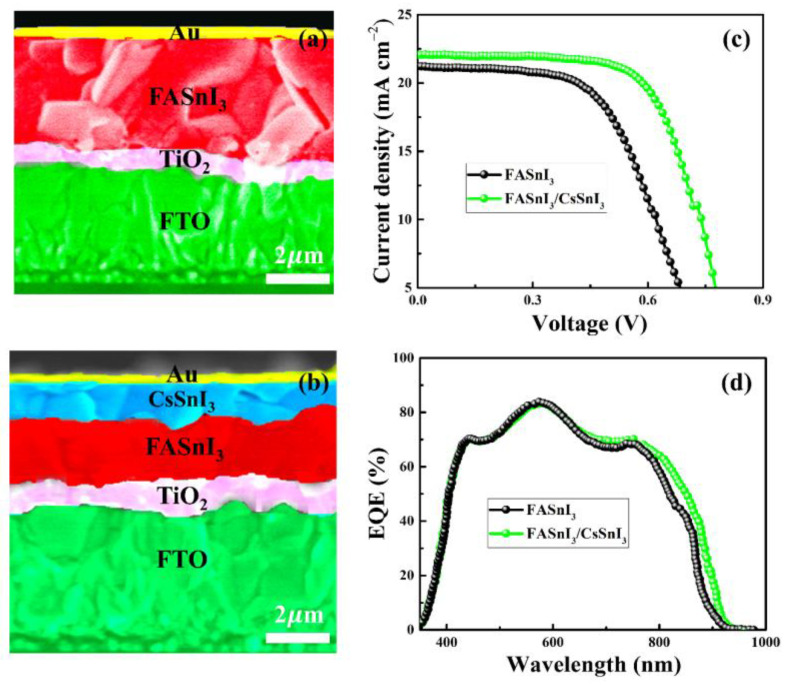
The cross-sectional SEM images (**a**,**b**), J-V characteristic curves (**c**), and EQEs of the HTM-free PSCs using FASnI_3_ and FASnI_3_/CsSnI_3_ as photo-absorber layers (**d**).

**Table 1 nanomaterials-13-00983-t001:** Material parameters for the homojunction-based HTM-free PSC. Here *L*, *Φ*, ε*_r_*, *E_g_*, *Χ*, *N_c_*, *N_v_*, *µ_n_*, *µ_p_*, *N_A_*, *N_D_*, and *N_t_* represent thickness, work-function, relative permittivity, energy band gap, electron affinity, effective conduction band density, effective valance band density, electron mobility, hole mobility, acceptor doping density, donor doping density, and defect density, respectively.

Parameters	FTO [56,57]	TiO_2_ [55,58,59]	FASnI_3_ [60,61,62]	IDL1 [63]	CsSnI_3_ [35,36,37,38]	IDL2 [64,65]	Au [66,67]
*L* (nm)	500	150	400	15	100	15	80
*Φ* (eV)	4.2	4.1	---	---	---	---	5.1
ε * _r_ *	9	10	25	10	25	10	---
*E_g_* (eV)	3.4	3.2	1.4	1.5	1.3	1.5	---
*Χ* (eV)	4	4	3.93	3.93	3.7	3.93	---
*N_c_* (cm^−3^)	2.3 × 10^18^	1.2 × 10^19^	2.5 × 10^19^	2.5 × 10^19^	1.4 × 10^19^	2.5 × 10^19^	---
*N_v_* (cm^−3^)	2.3 × 10^19^	1.2 × 10^20^	1.8 × 10^20^	2.5 × 10^20^	1.4 × 10^18^	2.5 × 10^20^	---
*µ_n_* (cm^−2^ V^−1^ s^−1^)	0.3	0.006	0.6	0.6	0.6	0.6	---
*µ_p_* (cm^−2^ V^−1^ s^−1^)	0.1	0.005	0.6	0.6	42	0.6	---
*N_A_* (cm^−3^)	0	0	1 × 10^18^	1 × 10^18^	1 × 10^17^	1 × 10^18^	---
*N_D_* (cm^−3^)	2 × 10^19^	5 × 10^19^	1 × 10^18^	1 × 10^18^	0	1 × 10^18^	---
*N_t_* (cm^−3^)	1 × 10^16^	1 × 10^17^	1 × 10^16^	1 × 10^18^	1 × 10^15^	1 × 10^18^	---

**Table 2 nanomaterials-13-00983-t002:** Photovoltaic parameters of the as-simulated homojunction-based HTM-free PSCs under illumination of AM1.5G with an intensity of 1000 W·m^−2^.

Device Design	V_oc_ (V)	J_sc_ (mA cm^−2^)	FF (%)	PCE (%)
FTO/TiO_2_/FASnI_3_/Au	0.66	26.07	76.37	14.62
FTO/TiO_2_/FASnI_3_/CsSnI_3_/Au	0.78	27.65	79.74	19.03

**Table 3 nanomaterials-13-00983-t003:** Photovoltaic parameters of the as-fabricated HTM-free PSCs under illumination of AM1.5G with an intensity of 1000 W·m^−2^.

Device Design	V_oc_ (V)	J_sc_ (mA cm^−2^)	FF (%)	PCE (%)
FTO/TiO_2_/FASnI_3_/Au	0.79	21.20	53.35	8.94
FTO/TiO_2_/FASnI_3_/CsSnI_3_/Au	0.84	22.06	63.50	11.77

## Data Availability

All the data presented in the manuscript can be obtained from the corresponding authors by reasonable request.

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
