# Peer review of "Lead-Free Perovskite Homojunction-Based HTM-Free Perovskite Solar Cells: Theoretical and Experimental Viewpoints"

_nanomaterials, 2023, doi:10.3390/nano13060983_

Round 1

Reviewer 1 Report

In the manuscript titled “Lead-free perovskite homojunction-based HTM-free perovskite solar cells: Theoretical and experimental viewpoints”, Sajid et al. performed the simulation and experimental design to pursue the optimal performance of CsSnI3/FASnI3 homojunction HTM-free perovskite solar cells. The authors demonstrate that the addition of a CsSnI3 layer can effectively enhance the charge extraction of the solar cells by generating an inter energy level between FASnI3 and Au. This work is interesting and fits the scope of Nanomaterials. I have a few questions and suggestions to improve the manuscript further:

1. The authors should provide a more detailed explanation of how they arrived at the optimal thickness of 100 nm. In Figure 3, the PCE doesn’t appear to saturate and keeps increasing with thickness. Additionally, the thickness has a weak impact on the energy level alignment, making it unclear why 100 nm is better than others.

2. The straight decay observed in the TRPL measurement in Figure 6g seems unusual. Should it follow an exponential decay function?

3. The resulting PCE of CsSnI3/FASnI3 homojunction solar cells is only 11.7%, significantly lower than the simulated PCE of 19.0%. The authors should discuss the potential reasons for the discrepancy and suggest ways to improve the PCE. This may help the development of homojunction HTM-free solar cells and broaden the impact of this work. In my opinion, one possible reason for the lower PCE could be the interfacial effect between the perovskite and Au electrode (ref. Advanced Materials 2022, 34, 2108616), which has been observed in several HTM-free perovskite solar cells.

4. Some labels are missing in the figure captions, such as Figure 6 and Figure 7.

Reviewer 2 Report

the work is interesting because it concerns the currently important field of photovoltaics related to perovskite solar cells; the authors rightly point out the obstacles to the commercial use of perskite cells on a large scale - low durability and toxic components (lead) and some problems with the transport layers for charge carriers; the search for a lead-free perovskite composition is one of the directions to overcome obstacles, however, it leads to a decrease in efficiency; similarly with the transport layer of holes - various attempts to make the transport layer are not as effective as the optimal one at higher costs; the submission presents theoretical (conventional) cell modeling and some experimental trials;

the theoretical model is very simplified, however, without the authors' own contribution,  using instead a commercial numerical simulator which is not able to properly consider many details - therefore, the results of this part of the analysis should be approached with caution; the experimental trial was also described rather modestly in the submission; conclusions should be thus more precise (the difference between simulation and experiment needs to be discussed due to deficiencies in simulation and/or experiment)

the paper can be published provided some revision made -- 1) some discussion of the fidelity of used simulator must be added, 2) various routes to balance the lowering of the perovskite cell efficiency decrease in the lead-free approach should be mentioned (e.g. Materials 2022, 15, 2254, Nano Energy 75 (2020) 104751); 

lower importance details:

the presentation requires revision - citations [1] and [2] are too modest in the Introduction to summarise  the current state-of-art in perovskite cells (cf. e.g. Materials 2022, 15, 2254. https://doi.org/10.3390/ma15062254 and bibliography there).

also, the lead-free approach should be supported by more citations, especially in light of the different possible ways to compensate of the associated efficiency loss with other techniques (again, cf. Materials 2022, 15, 2254 and Nano Energy 75 (2020) 104751 by Laska et al.). The linguistic verification is required and a proofread of the text -- for examples formulae in equations shoud be ended with point, etc. 

Reviewer 3 Report

The manuscript nanomaterials-2286845 titled Lead-free perovskite homojunction-based HTM-free perovskite solar cells: Theoretical and experimental viewpoints by Sajid Sajid et al, it is a nice study that may be well accepted by a broad community. However, requires revision prior to take any decision. Please, see the comments below.

Titled: It is ok.

Abstract: Ok but can you justify the low expected efficiencies when compared with other studies perovskites based? Is the lead?

Introduction: Here when addressing the topic, it is relevant to clearly identify the existing environmental bottlenecks and why the present study aims to solve the same. This is partially accomplished but missing the component related to the green aspects of the technology and of the materials used. Please see Nandy, S et al in Green economy and waste management: An inevitable plan for materials science, Feb 2022 | Feb 2022, PROGRESS IN NATURAL SCIENCE-MATERIALS INTERNATIONAL 32 (1).

Moreover, worth to discuss the role of the space charge and this could be really affected by the new proposed structure (see Panigrahi, S et al in Mapping the space charge carrier dynamics in plasmon-based perovskite solar cells, Sep 14 2019 | JOURNAL OF MATERIALS CHEMISTRY A, 7 (34), pp.19811-19819), namely, the ones associated with thermal losses or carriers thermalization, as well.

Compare present data with ones published to explain the differences please (see Menda, UD  et al in High-performance wide-bandgap perovskite solar cells fabricated in ambient high-humidity conditions, Oct 4 2021 | MATERIALS ADVANCES, 2 (19) , pp.6344-6355).

Finally, to support the discussion, worth to revisit the origin of the carrier’s recombination mechanism, Materials and Methods:

Simulation method, ok. But: how many structures and devices were tested? How reproducible ad reliable they are? What is the error associated when evaluating/testing? What are the environmental conditions in which the films were tested? Did you notice ageing effects?

Results and discussion

Overall, the simulation part of the discussion is very well conducted, but it is missing the scientific impact. Please, consider the comments made above to improve the discussion of the results answering to the set of points raised above. Please the values presented in tables 2 and 3 are unrealistic, and only possible via computer. What is the meaning to compare a microamp, for instance with a picoamp? Please use realist figures in the table, for the current, the voltage, the efficiency and the fill factor.

Conclusions:

Overall, very good. Missing information concerning potential problems arose with the stability and the impact of this new proposition for a sustainable energy process.

Figures: Are OK.

References: require updated.
